# Prevalence of hepatitis B virus infection and its associated factors among students in N'Djamena, Chad

Nalda Debsikréo[1,2]*, Birwé Léon Mankréo[1], Azoukalné Moukénet[1,3], Merwa Ouangkake[4], Nathan Mara[4], Ali Mahamat Moussa[3,5], Ndèye Coumba Toure-Kane[1,2], Françoise Lunel-Fabiani[6,7]

1 Cheikh Anta Diop University, Dakar, Senegal, 2 Institut de Recherche en Santé, de Surveillance Épidémiologique et de Formation, Dakar, Sénégal, 3 University of N'Djamena, N'Djamena, Chad, 4 Hôpital de la Paix, N'Djamena, Chad, 5 Centre Hospitalier Universitaire la Référence, N'Djamena, Chad, 6 Centre Hospitalier Universitaire Angers, BAT IBS-4 rue Larrey, Angers, France, 7 Laboratoire HIFIH, UFR Santé département Médecine, SFR 4208-UPRES EA3859, Université d'Angers, Angers Cedex 01, France

* ndaldaeliane@gmail.com

## Abstract

### Introduction

Infection by hepatitis B virus (HBV) is a major issue in public health. The prevalence of HBV in Chad is 12.4%, all age groups considered. Here, we aimed to determine the prevalence of HBV and its associated factors among university students in N'Djamena, the country's capital.

### Methods

A cross-sectional survey of students at either the University of N'djamena or Emi Koussi University was conducted from 3 to 23 July 2021. All participating students provided signed, informed consent and were included in the study consecutively. Blood samples were collected, and serum tested for hepatitis B surface antigen (HBsAg) using the Determine HBsAg rapid test kit, with confirmation of positive tests on an Abbott Architect i1000SR analyzer. Descriptive analysis and logistic regression were used to determine associations between the outcome variable and independent/covariate variables.

### Results

A total of 457 students with a median age of 24 years were included across different faculties. The prevalence of HBV infection was 14.87% (68/457). Most students (75%) were aged 25 years or less. Unprotected sex was reported by 64.9% of the students and multiple sexual partners by 53.6%. Furthermore, 45.7% of them reported having no knowledge of hepatitis B. Having an HBsAg-positive mother (AOR: 2.11), having a history of transcutaneous medical procedures (AOR: 2.97) and living with a family (AOR: 4.63) were significantly associated with HBV status. Age ≥26 years appeared as a protective factor (AOR = 0.41).

**Data Availability Statement:** All relevant data are within the manuscript and its Supporting Information files.

**Funding:** The author(s) received no specific funding for this work.

**Competing interests:** The authors have declared that no competing interests exist.

**Abbreviations:** Anti-HBs, Hepatitis B surface antibody; AOR, Adjusted odd ratio; CI, Confidence interval; COR, Crude odd ratio; HBsAg, Hepatitis B surface antigen; HBV, Hepatitis B virus.

## Conclusion

Our study detected a high, 14.87% prevalence of HBV infection among students in N'dja-mena, Chad, and shed light on its associated factors. HBV prevention strategies should include raising awareness among students, making full hepatitis vaccination mandatory before children begin school, promoting mass screening to identify and treat chronic HBV carriers and reduce transmission, and reducing the cost of vaccination.

## Introduction

Despite the availability of a prophylactic vaccine, hepatitis B remains a major public health problem. Approximately, 900,000 persons die from hepatitis B every year; as such, it is among the leading causes of mortality worldwide [1]. Its causative agent, the hepatitis B virus (HBV), is a partially circular DNA virus belonging to the *Hepadnaviridae* family. It is a small, enveloped virus responsible for both acute (sometimes fulminant) and chronic hepatitis. Moreover, HBV has oncogenic potential [2]. Chronic HBV infection is widespread [3] and its overall prevalence is 3.2% [4].

Although data on hepatitis B in Africa is scarce, its prevalence is among the highest in the world. More than 80 million people are infected with HBV in Africa [5], representing a prevalence of 6.1% [6]. Furthermore, a high prevalence of hepatitis B surface antigen (HBsAg), exceeding 8% in the general population, is observed in sub-Saharan Africa [7]. Hepatitis B is underdiagnosed on the continent, stemming from the frequently silent nature of chronic HBV infection and a lack of public information on its prevention, risks and consequences. Poor general health and vaccinal coverage also contribute to an increased risk of HBV infection in the population [8], with the risk of developing hepatocellular carcinoma [9].

Testing and access to treatment remain limited in many low-income countries, including those of Africa, despite the high burden of HBV infection frequently observed in them [10]. More than 124,000 Africans die each year from undetected and/or untreated hepatitis [11]. Considering these aspects, population screening for HBV infection is a major challenge along the path toward preventing, treating and potentially eliminating hepatitis B [12].The Global Burden of Disease Study 1990–2019 on hepatitis B reported an all-ages prevalence of 12.4% for the disease in Chad in 2019 [13]. This significant public health issue thus concerns Chad directly and requires active surveillance there. Infected individuals can transmit HBV to others via such biological fluids as blood and sexual secretions. In areas of high endemicity and in the perinatal period, this transmission of HBV may be vertical or horizontal [14]. Non-vertical transmission may occur through blood transfusions, blood exposure accidents for health professionals, the use of non-sterile equipment for tattooing, piercing, acupuncture or intravenous drug use, or unprotected sexual intercourse [15].

Among these latter aspects, unprotected sexual activity and substance use in particular may increase the likelihood of HBV infection among university students, who are generally young and apparently healthy [16]. In addition, a family history of HBV infection, sexual activity and socioeconomic conditions have been reported to be the main risk factors for HBV infection among students in Bangui [17] and asymptomatic adolescents in Nigeria [18].

In the 2000s and with the goal of preventing transmission of HBV during childhood, several sub-Saharan countries began adding the HBV vaccine to their Expanded Programs on Immunization. This includes Chad, where the vaccine was added to its national plan in 2008 [19]. Despite this policy, the prevalence of HBV infection continues to increase, especially in the

population over 15 years of age (incidence of 26.46% versus a national incidence of 13.62%) [20]. This aspect could hamper the attainment of National Health Plan targets to eliminate viral hepatitis, water-borne diseases and other communicable diseases by 2030 [21]. Studies on the burden of HBV infection among the student population and factors associated with hepatitis B are limited in this community, making it necessary to conduct studies to guide policies for the prevention of this scourge in Chad. Thus, we aimed to determine the prevalence of HBV infection and identify associated factors among a sampling of students in N'Djamena, the capital city of Chad.

## Methods and materials

### Study design

A cross-sectional study was conducted at the University of N'djamena (a public institution) and Emi Koussi University (a private institution, also located in N'djamena) from 3 to 23 July 2021. These institutions were selected because they have the largest number of students and N'Djamena is the city with the highest incidence of HBV (42.16%) in Chad [20]. The University of N'Djamena is the largest public university in the country, counting 16,142 students, seven faculties and four campuses [22]. Emi Koussi University is one of the largest private universities with 3,000 students approximately, two faculties and two campuses [23].

### Inclusion criteria

Lists of all faculties of the two universities were obtained from their respective administrations. Based on enrollment in them, students who accepted to participate in the study were enrolled consecutively irrespective of age, sex, other sociodemographic characteristics or risk factors.

### Sample size determination and sampling technique

A single-population sample size was calculated using the formula $n = \frac{z^2 pq}{i^2}$. Given that there is no similar study in a student environment in Chad, a prevalence of 50%, a z-score value of 1.96 with an alpha level of 0.05 (95% confidence) and a 10% relative precision were used. This yielded a minimum sample size of 422 students and we ultimately interviewed and collected samples from 457 students. The final sample size was distributed proportionally to the number of students in each faculty: Faculty of Languages, Letters, Arts and Communication (75 students); Faculty of Educational Sciences (80 students); Faculty of Exact and Applied Sciences (50 students); Faculty of Economics and Management (88 students); Faculty of Human and Social Sciences (75 students) and Emi Koussi University (50 students).

### Data collection method

A structured questionnaire was developed based on that of Rajamoorthy *et al*. [24]. It was used to collect information on socio-demographic characteristics and was administered during face-to-face meetings with the students. The questionnaire included age, sex, marital status, place of origin, faculty and study level, and risk factors, such as number of sexual partners, history of blood transfusion, surgery or hospitalization, HBsAg status of the mother, family history of HBV infection, HBsAg status of family members, history of traditional transcutaneous exposures such as piercings, tattoos, scarification, circumcision or excision (grouped as "risky practices"), history of transcutaneous medical procedures (including acupuncture), drug use, prison stays, unprotected sex, alcohol abuse, sharp object wounds, mode of residence, number of roommates, clothes sharing, and sleeping arrangements.

**Study variables.** In light of a literature review [25–27], the following variables were included in this study: sex, age, marital status, educational level and study site. With regard to risk factors, the following potential variables were retained: student's life history, including number of sexual partners, history of blood transfusion, surgery, hospitalization, maternal HBsAg status, family history of HBV infection, risky practices, transcutaneous medical procedures, drug use, prison stay, unprotected sex, alcohol, sharp injuries, number of roommates, living conditions, sharing of clothing, living in a family, history of hepatitis B testing.

## Specimen collection and processing

The study participants had a 5 ml venous blood sample drawn by a lab technician in an EDTA tube. The blood samples were labeled with unique numerical identifiers and coagulated. Sera were separated by centrifugation at 3000 rpm for 5 minutes and placed into Eppendorf tubes. HBsAg was searched for in the sera samples using the qualitative, rapid-detection immuno-chromatographic ALERE Determine HBsAg test, providing sensitivity and specificity of 95.3% and 93.3% respectively according to the manufacturer [28]. Positive results were confirmed on an Abbott Architect i1000SR analyzer (Abbott Diagnostics, Abbott Park, IL, USA) via chemi-luminescent microparticle immunoassay. After HBsAg screening, sera samples were frozen and stored at -80°C in the biobank laboratory of the national reference hospital.

## Data analysis

Descriptive analysis and logistic regression were used to examine associations between the outcome variable and independent/covariate variables. Quantitative variables were expressed as mean ± SD. Categorical variables were expressed as figures and percentages. The independent t-test was used to compare the means of continuous variables. When normal distribution or equal variance could not be assumed, the Mann-Whitney U test was used. The chi-squared test was used to compare proportions. For multivariate analysis, logistic regression analysis was performed to identify factors associated with HBV serological status among students. The regression was performed in two selection stages. The first consisted of automatic selection. The variables were selected according to the statistical selection criteria by a descending stepwise procedure. The model was selected by excluding the non-significant variables, using Akaike's information criterion as an index of parsimony. In this first stage of selection, five variables were selected, namely age, history of surgery, maternal HBV, transcutaneous medical procedures and type of housing. In the second stage, variables with a p-value <0.3 in univariate analysis were included in the initial multivariate logistic regression model [29]. Gender was included as a forced variable in the multivariate analysis, even though it had a p-value >0.3 in the univariate analysis. In this second stage, the variables marital status and injected-drug use were selected. Adjusted odds ratios (AORs) were calculated to test the statistical associations between the dependent and independent variables, after adjustment for the other selected variables. Those with a p-value <0.05 were considered statistically significant. A 95% confidence interval was calculated for all odds ratios. All analyses were performed using R version 4.1.2.

## Ethical considerations

The study was approved by the National Bioethics Committee of Chad (N° 201/PR/MESRI/DG/CNBT/2020, 8 November 2020) and the UCAD Research Ethics Committee (CER /UCAD/AD/MSN/050/2020, 7 December 2021). Written, informed and signed consent was obtained from all participants before inclusion in the study. The objectives of the survey were presented to the deans of the concerned faculties. All participants were informed of their serological results and provided with information on the possible consequences of HBV infection

and, if necessary, the measures to be taken. The subjects who tested positive for HBV were referred to the program "Fight Against Hepatitis in Chad" for advice and follow-up, if warranted.

## Results

### Socio-demographic characteristics

Of the 457 students who participated in the study, 348 (76%) were male and 109 (24%) female, i.e., a sex ratio (M/F) of 3.19. Their median age was 24 years and a majority (75%) were aged less than 25 years. The majority of the participants were from the University of N'djamena (407, 89%). Also, most were single (348, 76%) (**Table 1**).

### Prevalence of HBV and its associated risk factors

Of the 457 students participating in the study, 68 (14.87% (95% CI = 13.9–21.7%) were positive for HBsAg. Table 2 depicts the percentage of people interviewed who displayed risk factors and the prevalence of HBsAg among them. HBsAg positivity was highest among students living with a family (95.6%, n = 65), followed by students having unprotected sex (65%, n = 44), having multiple sexual partners (57%, n = 39), living in a density of residence of more than two (50%, n = 34), sharing sharp objects (46%, n = 31) and having an HBsAg-positive mother (34%, n = 23). Of the 107 (23%) students who had undergone a screening test for hepatitis B previously, 25 were HBsAg positive (37%) (Table 2).

### Bivariate logistic regression analysis of the factors associated with HBsAg positivity among students in N'djamena, Chad

Logistic regression was used to test the association between HBV infection and the independent variables and to identify risk factors. Considering a p-value <0.30 in bivariate analysis,

**Table 1. Socio-demographic characteristics of study participants.**

| Characteristic | Category | Frequency | Percent (%) |
|---|---|---|---|
| Gender | Female | 109 | 23.85 |
| | Male | 348 | 76.14 |
| Age | ≤25 | 343 | 75.05 |
| | >26 | 114 | 24.94 |
| study framework | FLLAC | 75 | 16.41 |
| | FES | 99 | 21.66 |
| | FEAS | 50 | 10.94 |
| | FEM | 100 | 21.88 |
| | FHSS | 83 | 18.13 |
| | EKU | 50 | 10.94 |
| Study level | Bachelor 1 | 141 | 30.85 |
| | Bachelor 2 | 143 | 31.29 |
| | Bachelor 3 | 168 | 36.76 |
| | Master & PhD | 5 | 1.09 |
| Marital status | Married | 109 | 23.85 |
| | Single | 348 | 76.14 |

FLLAC (Faculty of Languages, Letters, Arts and Communication); FES (Faculty of Educational Sciences); FEAS (Faculty of Exact and Applied Sciences); FEM (Faculty of Economics and Management); FHSS (Faculty of Humanities and Social Sciences); EKU (Emi-Koussi University).

**Table 2. Prevalence of HBV risk factors.**

| Characteristic | Overall n (%) | HBsAg positive n (%) | p-value |
|---|---|---|---|
| Knowledge of hepatitis B | 248 (54.26%) | 38 (55.9%) | 0.8 |
| History of blood transfusion | 58 (12.69%) | 10 (14.7%) | 0.6 |
| Multiple sexual partners | 252 (55.14%) | 39 57.4%) | 0.7 |
| History of hospital admission | 108 (26.63%) | 16 (23.5) | >0.9 |
| HBsAg-positive mother | 102 (22.31%) | 23 (33.8%) | 0.014 |
| Family history of hepatitis B | 71 (15.53%) | 15 (22.05%) | 0.3 |
| Transcutaneous medical procedures | 65 (14.22%) | 18 (26.5%) | 0.002 |
| History of drug injection | 15 (3.3%) | 1 (1.5%) | 0.7 |
| Risky practices (piercings, tattoos, etc.) | 136 (29.75%) | 21 (30.88%) | 0.8 |
| Jail stay | 28 (6.1%) | 5 (7.4%) | 0.6 |
| Unprotected sex | 297 (64.98%) | 44 (64.7%) | >0.9 |
| Sharing of sharp materials | 186 (40.70%) | 31 (45.6%) | 0.4 |
| Housing style: family | 353 (77.24%) | 65 (95.6%) | 0.004 |
| Density of residence: >2 | 245 (53.61%) | 34 (50. 00%) | 0.5 |
| History of hepatitis B testing | 107 (23.41%) | 25 (36.76%) | 0.005* |

having an HBsAg-positive mother, a history of transcutaneous medical procedures, age, marital status, a history of drug injection, housing style and a number of roommates were identified as candidate variables for multivariate analysis (Table 3).

**Multivariate logistic regression analysis of the of the factors associated with HBsAg positivity among students in N'djamena, Chad.** The odds of HBsAg positivity were found to be 2.03 times higher in students having an HBsAg-positive mother (AOR = 2.11, 95% CI: 1.16–3.78, p = 0.013) in comparison to students not having an HBsAg-positive mother. The odds of HBsAg positivity among students having a history of transcutaneous medical procedures (AOR = 2.97; 95% CI: 1.51–5.76, p = 0.001) was 3.14 times higher than students not having a history of transcutaneous medical procedures. The study showed that students living with a family (AOR = 4.63, 95% CI: 1.50–20.52, p = 0.018) were 4.54 times more likely to be HBsAg + in comparison to students living alone. In contrast, students aged ≥26 years had lower odds of HBsAg positivity (AOR = 0.41, 95% CI: 0.18–0.85, p = 0.022) in comparison to students aged ≤25 years (Table 4).

## Discussion

This study assessed the prevalence of HBV and its associated factors among students in the main public university and the main private university in N'Djamena, Chad. Its findings show a prevalence of HBV infection (HBsAg positivity) of 14.87% (68/457). That percentage places our study population in the high prevalence group, i.e., well above the 8% threshold established by the World Health Organization [30].

The HBsAg prevalence found in our study is also higher than the overall prevalence of 12.4% reported for all age groups in Chad, and over those reported for some other population groups in N'Djamena, for example 7.2% in pregnant women [31] and 13.5% in HIV-infected patients [32]. The higher HBsAg prevalence may be due to the fact that students were largely part of a relatively young but not young enough population. Indeed, with an average age of 24 ±3 years, our subjects most likely did not benefit from vaccination against HBV in childhood as part of the Expanded Program on Immunization introduced in 2008 [19]. Outside of that program, the relatively onerous price of the HBV vaccine is not covered by the current Chadian healthcare system, which acts as a barrier to large coverage in this population.

**Table 3. Bivariate logistic regression analysis of the factors associated with HBsAg positivity (n = 457).**

| Characteristics | Category | HBsAg-n (%) | HBsAg+n (%) | p-value | OR | [95% CI] |
|---|---|---|---|---|---|---|
| Sex | Female | 93 (23.9) | 16 (23.5) | | 1 | |
| | Male | 296 (76.1) | 52 (76.5) | 0.792 | 1.09 | 0.58–2.16 |
| Age | ≤ 25 | 287 (73.8) | 56 (82.4) | | 1 | |
| | ≥ 26 | 102 (26.2) | 12 (17.6) | 0.018* | 0.39 | 0.17–0.82 |
| Marital status | Married | 92 (23.7) | 17 (25.0) | | 1 | |
| | Single | 297 (76.3) | 51 (75.0) | 0.146* | 0.58 | 0.28–1.22 |
| History of surgical procedure | No | 367 (94.3) | 68 (100.0) | | 1 | |
| | Yes | 22 (5.7) | | 0.984 | 0.00 | 0.00–1597502.09 |
| History of blood transfusion | No | 341 (87.7) | 58 (85.3) | | 1 | |
| | Yes | 48 (12.3) | 10 (14.7) | 0.327 | 1.50 | 0.64–3.28 |
| Multiple sexual partners | >1 | 213 (54.8) | 39 (57.4) | | 1 | |
| | 1 | 176 (45.2) | 29 (42.6) | 0.896 | 0.96 | 0.53–1.73 |
| Risky practices (piercings, tattoos, etc.) | No | 274 (70.4) | 47 (69.1) | | 1 | |
| | Yes | 115 (29.5) | 21 (30.9) | 0.821 | 0.93 | 0.48–1.74 |
| History of hospital admission | No | 297 (76.3) | 52 (76.5) | | 1 | |
| | Yes | 92 (23.7) | 16 (23.5) | 0.830 | 1.07 | 0.55–2.03 |
| HBsAg-positive mother | No | 310 (79.7) | 45 (66.2) | | 1 | |
| | Yes | 79 (20.3) | 23 (33.8) | 0.024* | 2.02 | 1.09–3.71 |
| History of drug injection | No | 375 (96.4) | 67 (98.5) | | 1 | |
| | Yes | 14 (3.6) | 1 (1.5) | 0.237* | 0.27 | 0.01–1.60 |
| Jail stay | No | 366 (94.1) | 63 (92.6) | | 1 | |
| | Yes | 23 (5.9) | 5 (7.4) | 0.786 | 0.85 | 0.24–2.51 |
| Unprotected sex | No | 136 (35.0) | 24 (35.3) | | 1 | |
| | Yes | 253 (65.0) | 44 (64.7) | 0.859 | 0.95 | 0.52–1.76 |
| Transcutaneous medical procedures | No | 342 (87.9) | 50 (73.5) | | 1 | |
| | Yes | 47 (12.1) | 18 (26.5) | 0.001* | 3.13 | 1.53–6.31 |
| Sharing of sharp materials | No | 234 (60.2) | 37 (54.4) | | 1 | |
| | Yes | 155 (39.8) | 31 (45.6) | 0.337 | 1.32 | 0.75–2.31 |
| Housing style | Alone | 48 (12.3) | 3 (4.4) | | 1 | |
| | Family | 341 (87.7) | 65 (95.6) | 0.022* | 4.49 | 1.43–20.16 |
| Number of roommates | 1–2 | 178 (45.8) | 34 (50.0) | 0.19* | 1.48 | 0.82–2.65 |
| | >2 | 211 (54.2) | 34 (50.0) | | 1 | |

95% CI = 95% Confidence Interval, OR: odds ratio, 1: reference category, *: p<0.30: candidate variables for multivariate analysis.

Moreover, the HBV prevalence found in our population is higher than those reported in studies on students from Togo (4.6%) [26], Cameroon (5.6%) [33] and Ethiopia (11.5%) [34]. It is lower however than that reported in Nigeria (31.5%) [35] and comparable to that reported in the Central African Republic (15.5%) [17].

Currently, Chadian students are not provided with prevention information (e.g. on sexuality, sexually transmitted diseases and risky practices), which most likely plays a part in that high prevalence rate. The environment in which students live may also carry risk factors (presence of drugs, alcohol, sex, etc.) [16] and the students themselves are in a period of life where risk taking may be more common, as compared to other age groups. Furthermore, sex education continues to be a taboo subject in Chadian families.

Of the 107 students (23%) who had already been tested for HBV infection in the past, 25 (37%) tested positive in our study. That finding is consistent with data reported by Halatoko

**Table 4. Multivariate logistic regression analysis of the of the factors associated with HBsAg positivity (n = 457).**

| Characteristics | Category | HBsAg- n (%) | HBsAg+ n (%) | COR [95% CI] | P value | AOR [95% CI]) | P value |
|---|---|---|---|---|---|---|---|
| Sex | Female | 93 (23.9) | 16 (23.5) | - | | - | |
| | Male | 296 (76.1) | 52 (76.5) | 1.09 (0.58–2.16) | 0.792) | - | |
| Age | ≤ 25 | 287 (73.8) | 56 (82.4) | - | | - | |
| | ≥ 26 | 102 (26.2) | 12 (17.6) | 0.39 (0.17–0.82) | 0.018 | 0.41 (0.18–0.85) | 0.022* |
| Marital status | Married | 92 (23.7) | 17 (25.0) | - | | - | |
| | Single | 297 (76.3) | 51 (75.0) | 0.58 (0.28–1.22) | 0.146 | 0.65 (0.33–1.31) | 0.219 |
| History of surgical procedure | No | 367 (94.3) | 68 (100.0) | - | | - | |
| | Yes | 22 (5.7) | | 0.00 (0.00–1597502.09) | 0.984 | - | |
| History of blood transfusion | No | 341 (87.7) | 58 (85.3) | - | | - | |
| | Yes | 48 (12.3) | 10 (14.7) | 1.50 (0.64–3.28) | 0.327 | - | |
| Multiple sexual partners | >1 | 213 (54.8) | 39 (57.4) | - | | - | |
| | 1 | 176 (45.2) | 29 (42.6) | 0.96 (0.53–1.73) | 0.896 | - | |
| History of hospital admission | No | 297 (76.3) | 52 (76.5) | - | | - | |
| | Yes | 92 (23.7) | 16 (23.5) | 1.07 (0.55–2.03) | 0.830 | - | |
| HBsAg positive mother | No | 310 (79.7) | 45 (66.2) | - | | - | |
| | Yes | 79 (20.3) | 23 (33.8) | 2.02 (1.09–3.71) | 0.024 | 2.11 (1.16–3.78) | 0.013* |
| History of drug injection | No | 375 (96.4) | 67 (98.5) | - | | - | |
| | Yes | 14 (3.6) | 1 (1.5) | 0.27 (0.01–1.60) | 0.237 | 0.29 (0.02–1.59) | 0.244 |
| Jail stay | No | 366 (94.1) | 63 (92.6) | - | | - | |
| | Yes | 23 (5.9) | 5 (7.4) | 0.85 (0.24–2.51) | 0.786 | - | |
| Unprotected sex | No | 136 (35.0) | 24 (35.3) | - | | - | |
| | Yes | 253 (65.0) | 44 (64.7) | 0.95 (0.52–1.76) | 0.859 | - | |
| Transcutaneous medical procedures | No | 342 (87.9) | 50 (73.5) | - | | - | |
| | Yes | 47 (12.1) | 18 (26.5) | 3.13 (1.53–6.31) | 0.001 | 2.97 (1.51–5.76) | 0.001* |
| Sharing of sharp materials | No | 234 (60.2) | 37 (54.4) | - | | - | |
| | Yes | 155 (39.8) | 31 (45.6) | 1.32 (0.75–2.31) | 0.337 | - | |
| Housing style | Alone | 48 (12.3) | 3 (4.4) | - | | - | |
| | Family | 341 (87.7) | 65 (95.6) | 4.49 (1.43–20.16) | 0.022 | 4.63 (1.50–20.52) | 0.018* |
| Number of roommates | 1–2 | 178 (45.8) | 34 (50.0) | 1.48 (0.82–2.65) | 0.192 | 1.45 (0.81–2.58) | 0.210 |
| | >2 | 211 (54.2) | 34 (50.0) | - | | - | |

95% CI = 95% Confidence Interval, COR: crude odds ratio, AOR: adjusted odds ratio, 1: reference category

*: significantly associated factors p-value <0.05

*et al.* in Togo [27]. Our interpretation is that students who knew they were already infected might have been particularly tempted to participate in the study, hoping to receive treatment or to check their serological status.

Focusing on the management and clinical follow-up for HBsAg-positive students, and encouraging those who tested HBsAg and anti-HBc-antibody-negative to complete the vaccination course are both critical. All participants in our study received information on HBV infection and those who tested positive were referred to health facilities for follow-up.

Our data showed that maternal HBV status was significantly associated with HBsAg carriage among our students, likely explained by mother-to-child (vertical) transmission risks. Results similar to ours were reported in Gambia in 2007, where vertical transmission was responsible for 16% of chronic infections [36] https://www.zotero.org/google-docs/?OBBVqDand in Cameroon, where the risk of vertical transmission was substantial despite birth-dose vaccination [37]. These data confirm the hypothesis that in sub-Saharan Africa, a

highly endemic region for HBV, vertical transmission merits as much attention as its horizontal counterpart [38].

Our study reported a significantly higher risk of transmission among students living in families, with a seroprevalence of 95.6% (65/68) in this study group. These results corroborate those obtained in the Central African Republic [17], where intrafamilial transmission was reported to be associated with hepatitis B in students. This association is probably due to contact between HBV-infected family members [39], the exchange of sharp objects and the sharing of personal-use items (those causing skin and mucous-membrane lesions), and a lack of knowledge on the different modes of transmission of HBV. For these reasons, public awareness and systematic screening of those living with infected people are essential.

A history of transcutaneous medical procedures was found to be a risk factor for hepatitis B in this study. This could be due to nosocomial transmission of HBV (resulting from non-compliance with universal recommendations to prevent postoperative infections and cross-contamination of equipment and medical devices) [40].

According to the multivariate analysis, students aged 26 years or more were less likely to be infected with HBV than those aged 25 years or less, suggesting age as a protective factor. Similar results were reported by Tula and Iyoha in Nigeria [35], where participants aged 20–24 years were more exposed than other groups. It is possible that the older students had already been infected during childhood or adolescence but cleared that earlier HBV more or less quickly, and thus did not developed chronic infection. Unfortunately, we were unable to check for anti-HBc antibodies (Ab) to confirm this hypothesis. Similarly, we did not check for anti-HBs Ab, which could have given us an idea of the percentage of vaccinated students.

## Strengths and limitations

Our study adds to data regarding prevalence of hepatitis B virus infection in Chad. Such findings are expected to guide public health strategies for the diagnosis of HBV pathologies among students and the general population. Although our work involved a mixed population of students from both the University of N'djamena and Emi-Koussi University, it does have certain limitations. As underlined above, we were unable to carry out total anti-HBc Ab, anti-HBc IgM Ab and anti-HBs Ab testing. This would have allowed for a distinction between subjects who were acutely infected (with anti-HBc IgM Ab), chronic carriers, showing recovery from an acute or chronic infection (anti-HBc Ab positive) [41, 42], or immunized against HBV (anti-HBs Ab positive). The face-to-face nature of our interviews may represent a limitation and constraints in the study design may have introduced selection bias.

Logistic regression was adjusted for other variables but only ten were taken into account. This may represent a limit on the control of confounding factors.

## Conclusion and recommendations

Our study showed that the prevalence of HBV infection in students was higher than the overall prevalence in Chad. Having an HBsAg-positive mother, a history of transcutaneous medical procedures, living with a roommate, and living with a family were significantly associated with HBV status. In view of our results, HBV prevention and treatment policies should include: awareness campaigns to inform at-risk populations like students on HBV transmission pathways; HBV vaccination at birth; mandatory full hepatitis vaccination status before children start school; the promotion of mass screening to identify & treat chronic HBV carriers and reduce transmission; reduction of vaccination costs; and the promotion of free-of-charge HBV screening for pregnant women.

### Inclusivity in global research

Additional information regarding the ethical, cultural, and scientific considerations specific to inclusivity in global research is included in the Supporting Information (S1 Checklist).

## Supporting information

**S1 Checklist. Inclusivity in global research.**
(DOCX)

**S1 Questionnaire.**
(DOCX)

**S1 File.**
(DOCX)

**S1 Data.**
(XLSX)

## Acknowledgments

We thank the laboratory technicians at Hôpital de la Paix in N'djamena who took the blood samples and performed the various tests. We express our gratitude to the survey's data collectors and supervisors. We furthermore thank the deans of Cheikh Anta Diop University in Dakar for their guidance. We express our gratitude both to the student secretaries who facilitated participation in the study and particularly to the students who joined us for it.

## Author Contributions

**Conceptualization:** Nalda Debsikréo, Merwa Ouangkake, Françoise Lunel-Fabiani.

**Data curation:** Nalda Debsikréo, Birwé Léon Mankréo, Azoukalné Moukénet.

**Formal analysis:** Nalda Debsikréo, Birwé Léon Mankréo, Azoukalné Moukénet.

**Funding acquisition:** Nalda Debsikréo.

**Investigation:** Nalda Debsikréo, Merwa Ouangkake, Nathan Mara, Ali Mahamat Moussa.

**Methodology:** Nalda Debsikréo, Birwé Léon Mankréo, Azoukalné Moukénet, Ali Mahamat Moussa, Ndèye Coumba Toure-Kane, Françoise Lunel-Fabiani.

**Project administration:** Nalda Debsikréo, Merwa Ouangkake, Nathan Mara.

**Resources:** Nalda Debsikréo.

**Software:** Nalda Debsikréo, Birwé Léon Mankréo, Azoukalné Moukénet.

**Supervision:** Nalda Debsikréo, Merwa Ouangkake.

**Validation:** Nalda Debsikréo, Birwé Léon Mankréo, Azoukalné Moukénet, Merwa Ouangkake, Nathan Mara, Ali Mahamat Moussa, Ndèye Coumba Toure-Kane, Françoise Lunel-Fabiani.

**Visualization:** Nalda Debsikréo, Birwé Léon Mankréo, Azoukalné Moukénet.

**Writing – original draft:** Nalda Debsikréo.

**Writing – review & editing:** Nalda Debsikréo, Birwé Léon Mankréo, Azoukalné Moukénet, Merwa Ouangkake, Nathan Mara, Ali Mahamat Moussa, Ndèye Coumba Toure-Kane, Françoise Lunel-Fabiani.

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
