## [Decision Letter · Decision Letter 0]

31 Jan 2023

PONE-D-22-22361Prevalence of hepatitis B virus infection and associated risk factors among students in N'Djamena, ChadPLOS ONE

Dear Dr. Nalda,

Thank you for submitting your manuscript to PLOS ONE. After careful consideration, we feel that it has merit but does not fully meet PLOS ONE’s publication criteria as it currently stands. Therefore, we invite you to submit a revised version of the manuscript that addresses the points raised during the review process.

We look forward to receiving your revised manuscript.

Kind regards,

Zacharie Tsala Dimbuene, Ph.D.

Academic Editor

PLOS ONE

Journal Requirements:

Additional Editor Comments (if provided):

Dear authors,

I commend that the manuscript has some potentials; however, it requires much more work then it stands now. First, there are many typos and grammatical errors which makes the manuscript unpleasant to the readers. Second, the sampling design of the study is not well presented. For instance, p.5, lines 104-106, authors said that "random sampling technique was used". Thereafter, they stated that a 1st student was selected (how?) and then gradually, other volunteers were enrolled. This is far from "random sampling". When random sampling is used, selected participants are well known in advance even before the study starts. Third, prevalence of HBV is significantly higher in N'Djamena among students. However, the paper is unable to give a context-specific explanation of this interesting finding (un)willingly. Fourth, the analyses are less convincing and inappropriately presented. Fifth, the discussion is less engaging because the authors are not presenting their findings in a way helping the readers to get a clear sense of the message they are conveying. This section also contains lot of grammatical errors and false sentences. For instance, on page 12 &13, lines 228-232, there is no logic in the affirmation "This result corroborates the WHO study which highlighted that more than 124,000 Africans die each year from the consequences of undetected and untreated hepatitis" while the first 2 sentences were about the differences in HBV prevalence between lifetime tested and untested students. Sixth, another (unjustifiable) variable included in the study is "HBV of family members". Since HBV can only be detected via lab exams, authors should elaborate a bit more on how this variable was collected in the study. Seventh, how do the authors theoretically explain the inclusion of "housing style" and draw a number of conclusions regarding its association with HBV among students?

Reviewers' comments:

Reviewer's Responses to Questions

**Comments to the Author**

1. Is the manuscript technically sound, and do the data support the conclusions?

Reviewer #1: Partly

Reviewer #2: Partly

Reviewer #3: No

Reviewer #4: Yes

2. Has the statistical analysis been performed appropriately and rigorously? 

Reviewer #1: No

Reviewer #2: No

Reviewer #3: No

Reviewer #4: Yes

3. Have the authors made all data underlying the findings in their manuscript fully available?

Reviewer #1: Yes

Reviewer #2: Yes

Reviewer #3: No

Reviewer #4: No

4. Is the manuscript presented in an intelligible fashion and written in standard English?

Reviewer #1: No

Reviewer #2: No

Reviewer #3: Yes

Reviewer #4: Yes

5. Review Comments to the Author

Reviewer #1: The study covers a significant problem especially in Africa and other third world countries.

comments

Grammatical typo errors should not be overlooked

Page 5 Samplng

line 104-105: how are the students selected, are they self selected? Randomly selected, systematic random sampling? It is not clear.

Data collection method: how did you prepare the structured tool? Was it standard? Did you refere some other works (cite).

Who collected the data? How many data collectors, their profession…

Biological analysis

what is the sensitivity and specificity of the kit?

Table 2:

the table is long (more than one page), can break it down

last row: history of HBV? It is not clear how the test turns negative from previous positive test? Are they cured?

Line 198-201: it is not clear how the interpretation is made

Table 3: please revise the interpretation by considering the confidence interval

what do you mean by OR1 two times?

How did you deal with variables which does not fulfill the chi square criteria?

line 224: how does high prevalence imply improvement in awareness?

Discussion: please revise following the correct result interpretation

Reviewer #2: Dear Author,

I am wishing to acknowledge author's work entitled: ‘Prevalence of hepatitis B virus infection and associated risk factors among students in N'Djamena, Chad’. The results illustrated the prevalence of hepatitis B virus among students. Therefore, this study will be useful for future prevention and control of hepatitis infection through the knowledge of prevalence HBV and risk factors among students. The manuscript is more or less in line with the vision, objectives and instructions of PLoS One. Despite its potential, however, this manuscript contains major errors in data presentation and interpretation, grammatical errors which need to be corrected (see the attached file).

General Comments

1. Although what authors are reporting is not a novelty, they must however be commended for their efforts in reporting what is prevalence of HBV among students in their locality. But there are so many researches worldwide which were done by more advanced lab techniques like ELISA and molecular techniques (PCR).

2. Use of English language is poor in certain sections and would require a detailed revision.

3. Based on PLoS One authors’ guideline, figures must be uploaded separately in tif or other form.

Title

• The title was good. But why you removed the area name ‘Emi Koussi’ from the title? You talked more about this study area in the methods part.

Abstract

• The punctuation colon (:) in each subheading was not necessary.

• Line 17; read as ‘Viral hepatitis B (HBV) is a major public health threat’. It is better written as ‘Viral hepatitis B virus (HBV) is a major public health problem’. Here the word ‘threat’ is not the appropriate word.

• Line 20-21; ‘This study aims to determine the prevalence-------------’ It is better written as, ‘This study aimed to determine the prevalence--------------------’

• Line 23; ‘-------- conducted in July 2021 including 457 students randomly selected-------------’. Here a specific length of time should be mentioned. Similarly, which random sampling technique was used? Simple or systematic?

• Which specimen type you have processed? It should be described before you talk about the method.

• Line 26; ‘Descriptive analyzes and binary logistic regressions were used to examine the relationship between dependent variables and sociodemographic including other risk factors’. It is better you say, ‘Descriptive analysis and binary logistic regressions were used to examine the relationship between dependent variables and socio-demographic factors including other risk factors.’

• Among how many study participants the study conducted?

• In the result part, it is better you include both the numerator and denominator together with percentage in presenting prevalence, for example A/B (14.87%).

• Too short and shallow result presentation

• Your result completely contrasts the fact. Are you interpreting the result inversely? See the data, ‘Students aged between 26-35 years (0.032), who were born from HBsAg positive mothers (0.027), who lived with a roommate (0.025) or had a history of hepatitis (p=0.015) were less likely to contract HBV infection’. For example, A participant who lived with a roommate has a probability of acquiring HBV 4.5 times than those living lonely (OR= 5.42, 95% CI: 1.55-25.6; p= 0.025). The author better go through the analysis and correct mistakes that appeared.

• What is your baseline to say ‘high prevalence of HBV infection’?

• Which type of conclusion is it? ‘This study highlights some solutions to improve the situation of hepatitis B in Chad and suggests policies to eliminate viral hepatitis’.

• Additional key words:-HBsAg, risk factor

• It should be re-written and make adjustments based on the comments in the methods and results part that follows.

Introduction

• The introduction seems like literature review, so it is better modified.

• The word treat is not the appropriate word; please replace by other word.

• The introduction described the burden among clinically ill participants, not apparently healthy individuals (e.g. students). Therefore, the author better show the burden of HBV in similar or related study populations.

• There was no description about the microbiology of HBV.

• I do not think the importance of writing values for P-value, 95%CI and OR. For example, ‘Infants infected during the perinatal period 62 have 3.74 times (OR=3.74, 95% CI: 0.97–14.39) risk of becoming chronic carriers [11]’.

• Line 82-84 should be deleted. ‘Identifying these risk factors for HBsAg 83 carriage with appropriate management would reduce the disease burden. Knowledge of the HBV 84 infection risk factors would also guide policies aimed at reducing the HBV burden’.

Materials and Methods

• Line 93; the subheading should be ‘Sample size determination and sampling technique’

• About the sampling technique, please see the comment in the abstract

• The ‘selection of variables’ subheading better be replaced by ‘Socio-demographic data collection’

• The ‘Biological analysis’ also should be replaced by ‘Laboratory data processing’

• In the ethical clearance part, were all participants above the age of 18 years old? If not, Why did not you took assent from participants below the age of 18 years old?

• Your participants were apparently healthy individuals (students). There should be a mechanism in place to link those who need treatment with the healthcare system. How do you manage the psychological and potential social harm on HBV positives that do not need treatment? So ethical issues were not well described.

• Specimen collection and processing needs to be well written.

Results

• Your subheading ‘Sample characteristics’ was not the appropriate one. It is better you replace by, ‘ Socio-demographic characteristics’

• Based on your sample size determination your sample size was 422. Why you included 457 in the result part?

• You mix up the socio-demographic characteristics of participants with prevalence of HBV as well as risk factors. I recommend you to present, first the socio-demographic characteristics and then prevalence of HBV, finally risk factors for HBV.

• What is your base to classify the age group? Have you associated with possible risks of interest? It is better again re-group and perform the analysis.

• Is there any difference between the two subheadings? ‘Factors associated with HBV infection among students’ and ‘Prevalence and risk factors associated with HBsAg carriage’. For me it was not clear.

• It was difficult to understand/catch up the concepts of the tables. They lacked OR and 95%CI. Yet you presented in the text. From where you get it (OR and 95%CI)?

• Similarly, it is better you represent the crude odds ratio by COR and the adjusted odds ratio by AOR than simply by OR.

• All table titles should be complete in description, area and period of study.

Discussions

• Since the necessity of doing this work was already described in the introduction part. So here writing about it was not necessary. Therefore, delete the following. Line 206 ‘To inform policies aiming to reduce the burden of HBV in Chad and to achieve the objectives of the 207 national health plan 2016-2030, this study was designed to determine the prevalence of HBV and to 208 identify the HBV infection risk factors, among students at the Universities of N'djamena and Emi Koussi’.

• Authors tried to compare their findings with different reports from worldwide. They also better justify the reason for variation among results of different research findings with respect to theirs’ based on actual situation; most of the reason for variation mentioned in the discussion part were already intensively described by other authors.

• Most of the literatures used for comparisons were researches on patients. In this case these literatures were incomparable to this study.

Conclusions

• Normally, conclusion emanates from your findings of the research.

• Revise it and accommodate recommendation of community/government importance.

Reviewer #3: In the submitted manuscript “Prevalence of hepatitis B virus infection and associated risk factors among students in N'Djamena, Chad” (Manuscript Number: PONE-D-22-22361), the authors wrote an original research article describing the prevalence of HBsAg positivity and factors associated for being HBsAg-positivity among university students in N'Djamena, Chad.

HBV infection is still a major health problem worldwide, especially in Africa including Chad. In Chad, HBV screening is not systematic. Most of infected people are unaware of their status. This study aims to determine the prevalence of HBsAg positivity and to identify factors associated with being HBsAg-positivity. The authors conducted a cross-sectional study among 457 university students in N'Djamena, Chad in July 2021. HBsAg seroprevalence determined by the Determine HBsAg rapid test was 15%. After logistic regression analyses, several factors associated with being HBsAg-positivity were identified. The study results indicate that there is an urgent need to test HBV and provide vaccination in student population in Chad.

However, in my point-of-view, there are some major and minor points that need to be improved or clarified:

1. The “Title” and “Abstract” section,

a. Please consider to use the word “Associated factors” rather than the word “Risk factors” for this cross-sectional study (REF: Antay-Bedregal D, Camargo-Revello E, Alvarado GF. Associated factors vs risk factors in cross-sectional studies. Patient Prefer Adherence. 2015 Nov 13;9:1635-6. doi: 10.2147/PPA.S98023), and also for the entire article.

b. Texts in the results part of the abstract is confused.

• “Students aged between 26-35 years (0R=0.90, 95% CI: 0.42-1.80; p= 0.032), who were born from HBsAg positive mothers (OR=0.48, 95% CI: 0.08-3.84; p = 0.027), who lived with a roommate (OR= 5.42, 95% CI:1.55-25.6; p= 0.025) or with family (OR= 2.69, 95%CI: 0.90-11.7; p= 0.025) or had a history of hepatitis (OR= 0.46, 95% CI : 0.25-0.86, p=0.015) were less likely to contract HBV infection.” In general, if OR > 1 means “they were more likely to contract HBV infection, as compared to the reference”. In contrast, if OR < 1 means “they were less likely to contract HBV infection, as compared to the reference”

2. In the “Introduction” section,

a. In line 56, the authors should to specify the population of each study, for example, Ref# 7 was conducted in general population, Ref# 8 was conducted in HIV population.

b. Line 61-62, authors wrote “Infants infected during the perinatal period have 3.74 times (OR=3.74, 95% CI: 0.97–14.39) risk of becoming chronic carriers [11]”. This sentence may mislead the meaning of the origin article. Ref#11 indicated that “Children having intervals longer than 10 weeks between their first two HB vaccine doses were at greater risk of developing chronic HBV infection. Their risk was 3.74 times higher than that of those with intervals shorter than 10 weeks (95% CI = 0.97–14.39).

c. Line 62-65, “Parenteral transmission occurs through blood transfusion, blood exposure accidents among health professionals and use of non-sterile equipment for tattooing, piercing, acupuncture, and intravenous drug injections. These risk factors are the main causes and most frequent routes of transmission [12].” Currently, this sentence may not be all correct. Please also update the reference.

d. Line 68-71, “Indeed, students are generally young, in good apparent health and sexually active, which puts them at risk of HBV infection. In addition, they spend more time in a crowded environment (amphitheater, dwellings, etc.) and HBV transmission may occur via saliva and skin exudates [13]”. Ref #13 is not relevant to the sentence.

e. Line 79-80, “Moreover, few studies have been performed on the burden of liver disease in at-risk populations such as students in Chad.” Please add the references. Or the authors would like to mention that “Study on the burden of HBV infection among student population in Chad is still limited.”, Please clarify.

3. In “Material and Methods” section,

a. Line 95-96, “a prevalence of 50% (close to 42.12%, the incidence of HBV in N'djamena)” Please add the reference.

b. In part “Biological analysis” (line 131-140), it is already a well-known information, the paragraph should be removed and be referred to the manufacturer’s recommendation.

4. In “Results” section, I would like to suggest the authors to re-analyze the data again with an advice from statisticians.

a. In Table 1, superscripted number#1 does not make sense, as well as, the Table’s footnote (line 176-178)

b. Indeed, Table 1 should demonstrate “Baseline sociodemographic, socioeconomic and other characteristics of the study population”. No HBsAg results needed.

c. Table 2 should demonstrate “HBsAg seroprevalence among the study population, probably describe in detail according to the baseline characteristics”

d. Table 3 should demonstrate “Univariate and multivariable analysis of factors associated for HBsAg positivity among the study population”. The authors did not indicate the reference group (OR=1) in the table. Table 3 Indeed, table 2 can probably be merged with table 3.

e. Data with missing value should not be taken to bivariate analysis, univariate and multivariable logistic regression analyses.

f. Indeed, a 95% confidence interval for the odds ratio also provides a test of the null hypothesis that the odds ratio is 1 at the 5% significance level. If the confidence interval does not include 1, we reject H0 and conclude that the odds for the two groups are different; if the interval does include 1, the data do not provide enough evidence to distinguish the groups in this way. The authors could not conclude that “Students aged between 26-35 years were 1.11 times less likely to be HBsAg positive (0R= 0.90, 95% CI: 0.42-1.80; p= 0.032)” (Line 196). Also, it is happened with other variables such as students with HBV-uninfected mothers and who lived with HBsAg+ family.

g. There are many good samples available for this kind of analysis, for example:

• Bancha B, Kinfe AA, Chanko KP, Workie SB, Tadese T (2020) Prevalence of hepatitis B viruses and associated factors among pregnant women attending antenatal clinics in public hospitals of Wolaita Zone, South Ethiopia. PLoS ONE 15(5): e0232653.

• Argaw, B., Mihret, A., Aseffa, A. et al. Sero-prevalence of hepatitis B virus markers and associated factors among children in Hawassa City, southern Ethiopia. BMC Infect Dis 20, 528 (2020).

• Kassaw B, Abera N, Legesse T, Workineh A, Ambaw G. Sero-prevalence and associated factors of hepatitis B virus among pregnant women in Hawassa city public hospitals, Southern Ethiopia: Cross-sectional study design. SAGE Open Med. 2022 Dec 6;10:20503121221140778.

• Makuza, J.D., Rwema, J.O.T., Ntihabose, C.K. et al. Prevalence of hepatitis B surface antigen (HBsAg) positivity and its associated factors in Rwanda. BMC Infect Dis 19, 381 (2019).

• Etc.

5. In “Discussion” section,

a. Need more improvement after re-analysis.

b. The author compared their results with other studied which conducted in different population, likes in HIV-infected people, in hospitalized patients…

c. How the authors ensure on data of HBsAg-positive family members? Questionnaires survey may provide a wrong result.

Best regards,

Reviewer #4: Methods

1) Kindly describe the randomisation procedure. Was it done at the faculty level? or was the random number generation for the whole population.

Results

2) If the majority of students are less than 25, should your age categories be re-classified? If so, please consider it.

3) the only sexual history is single/many partners. Do you have any additional data?

4) The first table overall, HbsAg+ and HbsAg-; and second table HbsAg+, HbsAg-, and overall. Please be consistent in presentation of tables

5) How did you build univariate/multivariate models? You have only p value for each group. How come? Kindly explain.

Hope these comments are useful

6. PLOS authors have the option to publish the peer review history of their article (what does this mean?). If published, this will include your full peer review and any attached files.

Reviewer #1: No

Reviewer #2: No

Reviewer #3: No

Reviewer #4: No

---

## [Author Response · Author response to Decision Letter 0]

13 Apr 2023

RESPONSE TO REVIEWERS

Reviewer #1: The study covers a significant problem especially in Africa and other third world countries.

Comments

Grammatical typo errors should not be overlooked Page 5 Sampling

We thank reviewer 1 for comments. We answered to all comments and correct typo (p 5) or grammatical errors.

Line 104-105: how are the students selected, are they self-selected? Randomly selected, systematic random sampling? It is not clear.

Thanks for the comment, based on number of students in faculties of the two largest universities of N’Djamena, students who volunteered to participate in the study were enrolled according to consecutive sampling. Line: 121-123. 

Data collection method: how did you prepare the structured tool? Was it standard? Did you refere some other works (cite).

A structured questionnaire was used to collect data. Line: 135. 

Who collected the data? How many data collectors, their profession…

Laboratory technicians collected the data under the supervision of the hepatitis focal point in Chad. There were five collectors.

Biological analysis what is the sensitivity and specificity of the kit? 

The kit had a sensitivity of 99.7% and a specificity of (99.5%) Reference: Bottero and t al J Hepatol Performance of rapid tests for detection of HBsAg and anti-HBsAb in a large cohort, France 2013

Table 2: the table is long (more than one page), can break it down:

Thank you, we changed this table according to the other reviewers comments. Line: 255-256.

Last row: history of HBV? It is not clear how the test turns negative from previous positive test? Are they cured?

Thanks for the comment: we were meaning history of hepatitis, or jaundice, we have clarified. 347-348.

Line 198-201: it is not clear how the interpretation is made Participants 197 with HBV-uninfected mothers were 198 2.08 times less likely to be HBsAg positive (OR=0.48, 95% CI: 0.10-4.26; p= 0.027). Students who lived with roommates or family had respectively 5.41 times (OR=5.41, 95% CI: 1.55- 25.6; p= 0.025) and 2.69 times (OR=2.69, 95% CI: 0.90-11.7; p= 0.025) risk to be HBsAg carriers than those who lived alone. Students who had a no history of Hepatitis B screening test were 2.17 times more????? likely to be HBsAg positive (OR= 0.46, 95% CI0.25-0.86, p=0.015) (Table 3)

Thank you, we have clarified the results section in the new version. Line: 36-44

Table 3: please revise the interpretation by considering the confidence interval what do you mean by OR1 two times? How did you deal with variables which do not fulfill the chi square criteria?

The independent t-test was used to compare the means of continuous variables. When normal distribution or equality of variance could not be assumed, the Mann-Whitney U test was used. The comparison of proportions was assessed by means of the Chi-square test. Line: 175

line 224: how does high prevalence imply improvement in awareness? 

Thank you, we have changed the sentence “and removed “this result corroborates….

Discussion: please revise following the correct result interpretation

Thank you, we have revised the interpretation and comments. Line: 268-368.

Reviewer #2: 

Dear Author, I am wishing to acknowledge author's work entitled: ‘Prevalence of hepatitis B virus infection and associated risk factors among students in N'Djamena, Chad’. The results illustrated the prevalence of hepatitis B virus among students. Therefore, this study will be useful for future prevention and control of hepatitis infection through the knowledge of prevalence HBV and risk factors among students. The manuscript is more or less in line with the vision, objectives and instructions of PLoS One. Despite its potential, however, this manuscript contains major errors in data presentation and interpretation, grammatical errors which need to be corrected (see the attached file).

General Comments

1. Although what authors are reporting is not a novelty, they must however be commended for their efforts in reporting what is prevalence of HBV among students in their locality. But there are so many researches worldwide which were done by more advanced lab techniques like ELISA and molecular techniques (PCR).

We thank reviewer for comments, we carried out additional tests by ARCHITECT HBsAg Quantitative confirming the results. Line: 162-168

2. Use of English language is poor in certain sections and would require a detailed revision.

Thank you for raising the point about grammatical errors, we carried out copyediting of the manuscript to improve the quality of its English.

3. Based on PLoS One authors’ guideline, figures must be uploaded separately in tif or other form.

Thanks for reminding us about the respect of style requirement of PLOS One related to figures. However, we have not a figure to integrate into this paper.

Title

• The title was good. But why you removed the area name ‘Emi Koussi’ from the title? You talked more about this study area in the methods part.

Thank you, "Emi Koussi" is the name of one of the universities whereas the study area itself is N'djamena.

Abstract

• The punctuation colon (:) in each subheading was not necessary.

Thank you for the remark. We removed it accordingly.

• Line 17; read as ‘Viral hepatitis B (HBV) is a major public health threat’. It is better written as ‘Viral hepatitis B virus (HBV) is a major public health problem’. Here the word ‘threat’ is not the appropriate word.

Thank you for mentioning this. We modified accordingly in -- line 20; "Hepatitis B virus (HBV) is a major public health problem. Thank you, we changed the sentence according to your suggestion.

• Line 20-21; ‘This study aims to determine the prevalence-------------’ It is better written as, ‘This study aimed to determine the prevalence--------------------’

Thank you, we changed the sentence according to your suggestion Line: 24-25.

• Line 23; ‘-------- conducted in July 2021 including 457 students randomly selected-------------’. Here a specific length of time should be mentioned. Similarly, which random sampling technique was used? Simple or systematic?

Thank you for your comment; we reviewed the methodology section. Line 27-28 “ A cross-sectional survey of students at the Universities of N'Djamena and Emi Koussi was conducted from 3 to 23 July 2021. Students were included in the study using a consecutive sampling”. Line 28-29. Simple random sampling technique was used to calculate sample size. Participants were selected consecutively but based on the volunteering to participate into the study.

• Which specimen type you have processed? It should be described before you talk about the method.

• Thanks, the type of sample taken was serum. Line 29.

• Line 26; ‘Descriptive analyzes and binary logistic regressions were used to examine the relationship between dependent variables and sociodemographic including other risk factors’. It is better you say, ‘Descriptive analysis and binary logistic regressions were used to examine the relationship between dependent variables and socio-demographic factors including other risk factors.’

Thank you for mentioning this. We replaced the sentence as suggested: line170-171; ‘Descriptive analysis and binary logistic regressions were used to examine the relationship between dependent variables and socio-demographic factors including other risk factors. 

• Among how many study participants the study conducted?

Thank you, Line 127-134 “A minimum sample size of 422 students was calculated. Nevertheless, we have interviewed and collected sample from 457 students.” The University of N'Djamena has 16,142 students. line: 115-116

• In the result part, it is better you include both the numerator and denominator together with percentage in presenting prevalence, for example A/B (14.87%).

Thank you for this comment, however we are afraid the tables will be too clumpy, although we put, in the narrative, after percentage the numerator and denominator between bracket to help the reader to understand the underlining calculation of percentages and proportions.

• Too short and shallow result presentation 

Your result completely contrasts the fact. Are you interpreting the result inversely? See the data, ‘Students aged between 26-35 years (0.032), who were born from HBsAg positive mothers (0.027), who lived with a roommate (0.025) or had a history of hepatitis (p=0.015) were morelikely to contract HBV infection’. For example, A participant who lived with a roommate has a probability of acquiring HBV 4.5 times than those living lonely (OR= 5.42, 95% CI: 1.55-25.6; p= 0.025). The author better go through the analysis and correct mistakes that appeared.

“Students aged between 26-35 years (0.032), who were born from HBsAg positive mothers (0.027), who lived with a roommate (0.025) or had a history of hepatitis (p=0.015) were more likely to contract HBV infection’

Thank you, we reviewed the analysis accordingly and we changed the result section and add more details and modified the interpretation which was wrong. Students were more likely…line:238-243.

• What is your baseline to say ‘high prevalence of HBV infection’? 

Thank you for this remark. By the way, we would rather say that this prevalence, higher than 8%, places our study population in the high endemicity group according to the WHO classification. We changed the sentence in the manuscript Line: 272-273.

• Which type of conclusion is it? ‘This study highlights some solutions to improve the situation of hepatitis B in Chad and suggests policies to eliminate viral hepatitis’. : This study highlights that HBV prevalence is still high in young adults i.e students in Chad.Thus, It is time to improve national health policies in Chad like hepatitis B prevention measures such as HBs Ag screening in pregnant women, vaccination at birth, catch up vaccination in children and students; and also to provide tests for screening the population, and, finally improve the management and treatment of infected subjects.

Thank you, you are right this study does highlight that HBV prevalence is still high in young adults i.e. students in Chad. Thus, we changed the sentence to line: 45-51.

• Additional key words:-HBsAg, risk factor

Thank you we added the suggested key words. Line: 52

• It should be re-written and make adjustments based on the comments in the methods and results part that follows.

Thank you, we took this remark into account. Lin: 26-34.

Introduction

• The introduction seems like literature review, so it is better modified.

Thank you, we changed the introduction section, according to your suggestion Line 53-107.

• The word threat is not the appropriate word; please replace by other word.

Thank you, we changed the sentence according to your suggestion. Line 54.

• The introduction described the burden among clinically ill participants, not apparently healthy individuals (e.g. students). Therefore, the author better show the burden of HBV in similar or related study populations.

Thank you, we changed the introduction according to your suggestion, by changing some references. Line 91-94.

• There was no description about the microbiology of HBV.

Thank you, we added a paragraph about the microbiology of HBV according to your suggestion. LINE 55-58.

• I do not think the importance of writing values for P-value, 95%CI and OR. For example, ‘Infants infected during the perinatal period 62 have 3.74 times (OR=3.74, 95% CI: 0.97–14.39) risk of becoming chronic carriers [11]’.

Thank you, we rewrote this sentence.

• Line 82-84 should be deleted. ‘Identifying these risk factors for HBsAg 83 carriage with appropriate management would reduce the disease burden. Knowledge of the HBV 84 infection risk factors would also guide policies aimed at reducing the HBV burden’.

Thank you, we deleted the sentence.

Materials and Methods

• Line 93; the subheading should be ‘Sample size determination and sampling technique’

Thank you, we considered the title ‘Sample size determination and sampling technique’: line119.

• About the sampling technique, please see the comment in the abstract

Thanks, Students were included in the study using a consecutive sampling”. Line 28-29. Simple random sampling technique was used to calculate sample size. Participants were selected consecutively but based on the volunteering to participate into the study.

• The ‘selection of variables’ subheading better be replaced by ‘Socio-demographic data collection’ 

Thank you, we considered the title ‘Socio-demographic data collection’: line 143.

• The ‘Biological analysis’ also should be replaced by ‘Laboratory data processing’

Thank you, we considered the title ‘Laboratory data processing’: line 153.

• In the ethical clearance part, were all participants above the age of 18 years old? If not, why did not you took assent from participants below the age of 18 years old? 

 Thank you, all participants were above the age of 18 years old. Age ranged from 19 to 59 years. Line 209.

• Your participants were apparently healthy individuals (students). There should be a mechanism in place to link those who need treatment with the healthcare system. 

HBsAg positive students were managed by the hepatitis B focal point line: 199-201.

How do you manage the psychological and potential social harm on HBV positives that do not need treatment? So ethical issues were not well described.

Thank you for raising this point, HBV-positive people were referred to the focal point of the program “fight against hepatitis in Chad” for follow-up and advice. line 199-201.

• Specimen collection and processing needs to be well written.

After HBs Ag screening test, sera samples were frozen and kept at …°C in the…laboratory.

After HBs Ag screening test, serum samples were frozen and kept at -80°C in the biobank laboratory of the national reference hospital Line: 166-167.

Results

• Your subheading ‘Sample characteristics’ was not the appropriate one. It is better you replace by, ‘Socio-demographic characteristics.

• Thank you, we considered the title ‘Socio-demographic characteristics’: line 207.

• Based on your sample size determination your sample size was 422. Why you included 457 in the result part?

Thank for raising this point. We provided details to ease the understanding of the readers Line : 127-134 “A minimum sample size of 422 students was calculated. Nevertheless, we interviewed and collected samples from 457 students”. 

• You mix up the socio-demographic characteristics of participants with prevalence of HBV as well as risk factors. I recommend you present, first the socio-demographic characteristics and then prevalence of HBV, finally risk factors for HBV.

Thank you, we took this comment into account. the socio-demographic characteristics line: 207, prevalence of HBV line: 234 and Factors associated with HBV infection among students line: 244.

• What is your base to classify the age group? Have you associated with possible risks of interest? It is better again re-group and perform the analysis.

Thank you, we reclassified the Age category: <20, 20 – 30, 30- 40 and > 40, referring to the article “bdoul Rahamane Njigou Mawouma, Amadou Hapsatou Djoulatou, Eliane Ornella Komnang, Etienne Omolomo Kimessoukie. Facteurs associés à l´infection de l´hépatite B chez les femmes enceintes dans les formations sanitaires du district de santé de Mokolo/Région de l´Extrême-Nord Cameroun. PAMJ. 21 Jan 2022. 41(61)”

• Is there any difference between the two subheadings? ‘Factors associated with HBV infection among students’ and ‘Prevalence and risk factors associated with HBsAg carriage’. For me it was not clear.

Thank you, we changed the titles. Line: 1

• It was difficult to understand/catch up the concepts of the tables. They lacked OR and 95%CI. Yet you presented in the text. From where you get it (OR and 95%CI)?

Thank you, we improved the reading of the tables and bring out the odds ratios and their 95% confidence intervals. Line 263-264.

• Similarly, it is better you represent the crude odds ratio by COR and the adjusted odds ratio by AOR than simply by OR.

Thank you, we took into consideration your suggestion, we represented the crude odds ratio by COR and the adjusted odds ratio by AOR. Line263-264.

• All table titles should be complete in description, area, and period of study.

Thank you for this advice, we took them into account: Table 1: Socio-demographic characteristics of study participants (line219-220), Table 2: Estimated prevalence of HBsAg (line 255-256) and Table 3: Factors associated with HBV status of study participants (line263-264).

Discussions

• Since the necessity of doing this work was already described in the introduction part. So here writing about it was not necessary. Therefore, delete the following. Line 206 ‘To inform policies aiming to reduce the burden of HBV in Chad and to achieve the objectives of the 207 national health plan 2016-2030, this study was designed to determine the prevalence of HBV and to 208 identify the HBV infection risk factors, among students at the Universities of N'djamena and Emi Koussi’.

Thank you, we deleted these sentences.

• Authors tried to compare their findings with different reports from worldwide. They also better justify the reason for variation among results of different research findings with respect to theirs’ based on actual situation; most of the reason for variation mentioned in the discussion part were already intensively described by other authors.

Thank you, we improved the discussion accordingly. Line: 268-357

Discussion

• Most of the literatures used for comparisons were researches on patients. In this case these literatures were incomparable to this study.

Thank you, we rewrote the discussion: line268-357.

Conclusions

• Normally, conclusion emanates from your findings of the research.

Thank you for the remark, the conclusion has been rewritten. line: 368-378.

• Revise it and accommodate recommendation of community/government importance.

Thank you for the remark, the recommendation has been rewritten. Line: 379-387.

Reviewer #3: 

In the submitted manuscript “Prevalence of hepatitis B virus infection and associated risk factors among students in N'Djamena, Chad” (Manuscript Number: PONE-D-22-22361), the authors wrote an original research article describing the prevalence of HBsAg positivity and factors associated for being HBsAg-positivity among university students in N'Djamena, Chad.

HBV infection is still a major health problem worldwide, especially in Africa including Chad. In Chad, HBV screening is not systematic. Most of infected people are unaware of their status. This study aims to determine the prevalence of HBsAg positivity and to identify factors associated with being HBsAg-positivity. The authors conducted a cross-sectional study among 457 university students in N'Djamena, Chad in July 2021. HBsAg seroprevalence determined by the Determine HBsAg rapid test was 15%. After logistic regression analyses, several factors associated with being HBsAg-positivity were identified. The study results indicate that there is an urgent need to test HBV and provide vaccination in student population in Chad.

However, in my point-of-view, there are some major and minor points that need to be improved or clarified:

1. The “Title” and “Abstract” section,

a. Please consider to use the word “Associated factors” rather than the word “Risk factors” for this cross-sectional study (REF: Antay-Bedregal D, Camargo-Revello E, Alvarado GF. Associated factors vs risk factors in cross-sectional studies. Patient Prefer Adherence. 2015 Nov 13;9:1635-6. doi: 10.2147/PPA.S98023), and also for the entire article.

Thank you for highlighting this comment, we have considered "associated factors".

b. Texts in the results part of the abstract is confused.

• “Students aged between 26-35 years (0R=0.90, 95% CI: 0.42-1.80; p= 0.032), who were born from HBsAg positive mothers (OR=0.48, 95% CI: 0.08-3.84; p = 0.027), who lived with a roommate (OR= 5.42, 95% CI:1.55-25.6; p= 0.025) or with family (OR= 2.69, 95%CI: 0.90-11.7; p= 0.025) or had a history of hepatitis (OR= 0.46, 95% CI : 0.25-0.86, p=0.015) were less likely to contract HBV infection.” In general, if OR > 1 means “they were more likely to contract HBV infection, as compared to the reference”. In contrast, if OR < 1 means “they were less likely to contract HBV infection, as compared to the reference”

Thank you, the result section has been changed. Line: 36-44.

2. In the “Introduction” section,

a. In line 56, the authors should specify the population of each study, for example, Ref# 7 was conducted in general population, Ref# 8 was conducted in HIV population.

Thank you, we specified the study population. Line 77-79.

b. Line 61-62, authors wrote “Infants infected during the perinatal period have 3.74 times (OR=3.74, 95% CI: 0.97–14.39) risk of becoming chronic carriers [11]”. This sentence may mislead the meaning of the origin article. Ref#11 indicated that “Children having intervals longer than 10 weeks between their first two HB vaccine doses were at greater risk of developing chronic HBV infection. Their risk was 3.74 times higher than that of those with intervals shorter than 10 weeks (95% CI = 0.97–14.39).

Thanks, these sentences have been deleted.

c. Line 62-65, “Parenteral transmission occurs through blood transfusion, blood exposure accidents among health professionals and use of non-sterile equipment for tattooing, piercing, acupuncture, and intravenous drug injections. These risk factors are the main causes and most frequent routes of transmission [12].” Currently, this sentence may not be all correct. Please also update the reference.

Thanks, we updated the reference.

d. Line 68-71, “Indeed, students are generally young, in good apparent health and sexually active, which puts them at risk of HBV infection. In addition, they spend more time in a crowded environment (amphitheater, dwellings, etc.) and HBV transmission may occur via saliva and skin exudates [13]”. Ref #13 is not relevant to the sentence.

Thanks, the reference "13" was deleted.

e. Line 79-80, “Moreover, few studies have been performed on the burden of liver disease in at-risk populations such as students in Chad.” Please add the references. Or the authors would like to mention that “Study on the burden of HBV infection among student population in Chad is still limited.”, Please clarify.

We wanted in fact to mention that “There is no studies on the burden of HBV infection among students’ population in Chad.” Line: 103-104.

3. In “Material and Methods” section,

a. Line 95-96, “a prevalence of 50% (close to 42.12%, the incidence of HBV in N'djamena)” Please add the reference.

Thanks, this sentence was deleted.

b. In part “Biological analysis” (line 131-140), it is already a well-known information, the paragraph should be removed and be referred to the manufacturer’s recommendation.

Thanks for the remark, the paragraph has been deleted and be referred to the manufacturer’s recommendation. Line: 161.

4. In “Results” section, I would like to suggest the authors to re-analyze the data again with an advice from statisticians.

a. In Table 1, superscripted number#1 does not make sense, as well as the Table’s footnote (line 176-178)

Thanks, superscripted number#1 and the Table’s footnote were deleted.

b. Indeed, Table 1 should demonstrate “Baseline sociodemographic, socioeconomic and other characteristics of the study population”. No HBsAg results needed.

Thank you, we took this comment into account in the new version. Line: 219-220.

c. Table 2 should demonstrate “HBsAg seroprevalence among the study population, probably describe in detail according to the baseline characteristics.”

Thank you, we took this comment into account; Line: 255-256.

d. Table 3 should demonstrate “Univariate and multivariable analysis of factors associated for HBsAg positivity among the study population”. The authors did not indicate the reference group (OR=1) in the table. Table 3 Indeed, table 2 can probably be merged with table 3.

Thank you, we cannot merge table 2 and table 3 because table 2 talks about prevalence and table 3 about logistic regression.

C. Data with missing value should not be taken to bivariate analysis, univariate and multivariable logistic regression analyses.

Thank you for the suggestion.

D. Indeed, a 95% confidence interval for the odds ratio also provides a test of the null hypothesis that the odds ratio is 1 at the 5% significance level. If the confidence interval does not include 1, we reject H0 and conclude that the odds for the two groups are different; if the interval does include 1, the data do not provide enough evidence to distinguish the groups in this way. The authors could not conclude that “Students aged between 26-35 years were 1.11 times less likely to be HBsAg positive (0R= 0.90, 95% CI: 0.42-1.80; p= 0.032)” (Line 196). Also, it is happened with other variables such as students with HBV-uninfected mothers and who lived with HBsAg+ family.

Thanks, we performed the analysis again. Line: 249-254.

g. There are many good samples available for this kind of analysis, for example:

• Bancha B, Kinfe AA, Chanko KP, Workie SB, Tadese T (2020) Prevalence of hepatitis B viruses and associated factors among pregnant women attending antenatal clinics in public hospitals of Wolaita Zone, South Ethiopia. PLoS ONE 15(5): e0232653.

Thank you, we followed this model.

• Argaw, B., Mihret, A., Aseffa, A. et al. Sero-prevalence of hepatitis B virus markers and associated factors among children in Hawassa City, southern Ethiopia. BMC Infect Dis 20, 528 (2020).

• Kassaw B, Abera N, Legesse T, Workineh A, Ambaw G. Sero-prevalence and associated factors of hepatitis B virus among pregnant women in Hawassa city public hospitals, Southern Ethiopia: Cross-sectional study design. SAGE Open Med. 2022 Dec 6;10:20503121221140778.

• Makuza, J.D., Rwema, J.O.T., Ntihabose, C.K. et al. Prevalence of hepatitis B surface antigen (HBsAg) positivity and its associated factors in Rwanda. BMC Infect Dis 19, 381 (2019).

• Etc.

5. In “Discussion” section,

a. Need more improvement after re-analysis.

The discussion has been improved after the new analysis Line: 268-357.

b. The author compared their results with other studied which conducted in different population, likes in HIV-infected people, in hospitalized patients…

We modified this section. Line: 277-278.

c. How the authors ensure on data of HBsAg-positive family members? Questionnaires survey may provide a wrong result.

We have based our work on Questionnaires survey.

Reviewer #4:

 Methods

1) Kindly describe the randomization procedure. Was it done at the faculty level? or was the random number generation for the whole population.

Thanks for this comment, A list of all faculties of the two largest universities of N’Djamena were obtained from its respective administrations. Based on number of students in faculties of the two largest universities of N’Djamena, students who volunteered to participate in the study were enrolled according to consecutive sampling. It does not concern the whole population.

Results

2) If the majority of students are less than 25, should your age categories be re-classified? If so, please consider it.

Thank you, we reclassified the Age category: <20, 20 – 30, 30- 40 and > 40. referring to the article “bdoul Rahamane Njigou Mawouma, Amadou Hapsatou Djoulatou, Eliane Ornella Komnang, Etienne Omolomo Kimessoukie. Facteurs associés à l´infection de l´hépatite B chez les femmes enceintes dans les formations sanitaires du district de santé de Mokolo/Région de l´Extrême-Nord Cameroun. PAMJ. 21 Jan 2022. 41(61)”

3) the only sexual history is single/many partners. Do you have any additional data?

Thank you, we also have the data on unprotected sex : line 226-227.

4) The first table overall, HbsAg+ and HbsAg-; and second table HbsAg+, HbsAg-, and overall. Please be consistent in presentation of tables.

Thank you, we have redone the table1. Line: 219-220.

5) How did you build univariate/multivariate models? You have only p value for each group. How come? Kindly explain.

Thank you, please see the methods section.181-187.

---

## [Decision Letter · Decision Letter 1]

24 Jul 2023

PONE-D-22-22361R1Prévalence et facteurs associés de l'infection par le virus de l'hépatite B chez les étudiants à N'Djamena, TchadPLOS ONE

Dear Dr. DEBSIKREO, 

Thank you for submitting your manuscript to PLOS ONE. After careful consideration, we feel that it has merit but does not fully meet PLOS ONE’s publication criteria as it currently stands. Therefore, we invite you to submit a revised version of the manuscript that addresses the points raised during the review process.

Please submit your revised manuscript by Sep 07 2023 11:59PM. If you will need more time than this to complete your revisions, please reply to this message or contact the journal office at plosone@plos.org. Please include the following items when submitting your revised manuscript:A rebuttal letter that responds to each point raised by the academic editor and reviewer(s). You should upload this letter as a separate file labeled 'Response to Reviewers'.A marked-up copy of your manuscript that highlights changes made to the original version. You should upload this as a separate file labeled 'Revised Manuscript with Track Changes'.An unmarked version of your revised paper without tracked changes. You should upload this as a separate file labeled 'Manuscript'.

We look forward to receiving your revised manuscript.

Kind regards,

Zacharie Tsala Dimbuene, Ph.D.

Academic Editor

PLOS ONE

**Additional Editor Comments:**

Dear Authors,

I commend that the manuscript revision has substantively improved. However, there still is a long way to go before the paper reaches the PLOS One standards. First, the introduction is flawed since there unclear sentences and not very sound currently (see annotated manuscript). Second, Tables 1 and 2 for instance has small numbers and this poses major issues for multivariate analyses because this could yield to unstable models. Third, there are several mistakes regarding references and sentences without verbs. Fourth, it is unclear whether the sample contains ALL students from N'Djamena (that what I guess later on) however, this should be made clear in the sampling design. Also, it is unclear about the allocation of the sample among universities.

Based on my own reading, I suggest the manuscript to be revised: MAJOR REVISION

Reviewers' comments:

Reviewer's Responses to Questions

**Comments to the Author**

1. If the authors have adequately addressed your comments raised in a previous round of review and you feel that this manuscript is now acceptable for publication, you may indicate that here to bypass the “Comments to the Author” section, enter your conflict of interest statement in the “Confidential to Editor” section, and submit your "Accept" recommendation.

Reviewer #5: (No Response)

Reviewer #6: All comments have been addressed

2. Is the manuscript technically sound, and do the data support the conclusions?

Reviewer #5: No

Reviewer #6: Yes

3. Has the statistical analysis been performed appropriately and rigorously? 

Reviewer #5: No

Reviewer #6: Yes

4. Have the authors made all data underlying the findings in their manuscript fully available?

Reviewer #5: No

Reviewer #6: Yes

5. Is the manuscript presented in an intelligible fashion and written in standard English?

Reviewer #5: No

Reviewer #6: Yes

6. Review Comments to the Author

Reviewer #5: This paper addressed an interesting topic which is a global public health issue, particularly in low income countries. So, I would like to acknowledge the authors for their effort to come up with this relevant topic. However this paper need a through edition and clarification to be an additional evidence for the scientific community and the general public health improvement.

Reviewer #6: I congratulate the authors for painstakingly making very rigorous revisions to the paper!

I will suggest a very minor edit:

Background in abstract and introduction: The word “systematic” attached to HBV screening should better read “not routinely done”.

Table 1: The class intervals for Age has some overlapping numbers, e.g., 20-30; 30-40. This should be corrected and “Ans” should be replaced with “years”

7. PLOS authors have the option to publish the peer review history of their article (what does this mean?). If published, this will include your full peer review and any attached files.

Reviewer #5: No

Reviewer #6: **Yes: **Jonah Musa

---

## [Author Response · Author response to Decision Letter 1]

7 Sep 2023

Reviewer's Responses to Questions

Comments to the Author

Reviewer #5: This paper addressed an interesting topic which is a global public health issue, particularly in low income countries. So, I would like to acknowledge the authors for their effort to come up with this relevant topic. However this paper need a through edition and clarification to be an additional evidence for the scientific community and the general public health improvement.

Our response:

Thanks for this comment: We have clarified the situation (see response to reviewers).

Reviewer #6: I congratulate the authors for painstakingly making very rigorous revisions to the paper!

I will suggest a very minor edit:

Background in abstract and introduction: The word “systematic” attached to HBV screening should better read “not routinely done”.

Table 1: The class intervals for Age has some overlapping numbers, e.g., 20-30; 30-40. This should be corrected and “Ans” should be replaced with “years”

 Our response:

Thank you for mentioning them: We have considered this comment.

---

## [Decision Letter · Decision Letter 2]

24 Oct 2023

PONE-D-22-22361R2Prevalence of hepatitis B virus infection and its associated factors among students in N'Djamena, ChadPLOS ONE

Dear Dr. DEBSIKREO,

Thank you for submitting your manuscript to PLOS ONE. After careful consideration, we feel that it has merit but does not fully meet PLOS ONE’s publication criteria as it currently stands. Therefore, we invite you to submit a revised version of the manuscript that addresses the points raised during the review process.

ACADEMIC EDITOR: The authors need to address/clarify the comments raised by reviewer #2 (see reviewers' comments) ==============================

We look forward to receiving your revised manuscript.

Kind regards,

Zacharie Tsala Dimbuene, Ph.D.

Academic Editor

PLOS ONE

Journal Requirements:

Reviewers' comments:

Reviewer's Responses to Questions

**Comments to the Author**

1. If the authors have adequately addressed your comments raised in a previous round of review and you feel that this manuscript is now acceptable for publication, you may indicate that here to bypass the “Comments to the Author” section, enter your conflict of interest statement in the “Confidential to Editor” section, and submit your "Accept" recommendation.

Reviewer #6: All comments have been addressed

Reviewer #7: All comments have been addressed

2. Is the manuscript technically sound, and do the data support the conclusions?

Reviewer #6: Yes

Reviewer #7: Yes

3. Has the statistical analysis been performed appropriately and rigorously? 

Reviewer #6: Yes

Reviewer #7: Yes

4. Have the authors made all data underlying the findings in their manuscript fully available?

Reviewer #6: Yes

Reviewer #7: Yes

5. Is the manuscript presented in an intelligible fashion and written in standard English?

Reviewer #6: Yes

Reviewer #7: Yes

6. Review Comments to the Author

Reviewer #6: I congratulate the authors for doing a thorough revision, and the responses are satisfactory. I recommend this version be accepted for publication after editorial checks and formatting.

Reviewer #7: In the submitted manuscript “Prevalence of hepatitis B virus infection and its associated factors among students in N’Djamena, Chad”, PONE-D-22-22361R2, this study addressed the high prevalence of HBV infection among university students in Chad. HBV prevention, especially HBV immunization, should be considered in low-income countries.

However, there are some points that need to be clarified:

1. The authors should mention inclusion criteria in the method part (e.g., sex, age, health status, etc.).

2. The data was collected using a questionnaire, was it a self-administered questionnaire or face-to-face interview? Please specify in data collection method.

3. The prevalence of HBV infection was 14.87%, what’s about 95%CI? please specify.

4. I would suggest using median age instead of mean age.

5. Considering p<0.30 in bivariate analysis, why did the authors select only five variables for multivariable logistic regression? (What about age, marital status, knowledge of HBV, history of drug injection, density of residence? there were also p<0.30.)

6. The authors did not ask about the history of HB immunization in the study participants. Although most likely did not benefit from vaccination against HBV in childhood as part of EPI, some may receive the vaccine in adult. We therefore can distinguish between who has protective or susceptible for HBV infection.

7. PLOS authors have the option to publish the peer review history of their article (what does this mean?). If published, this will include your full peer review and any attached files.

Reviewer #6: **Yes: **Jonah Musa

Reviewer #7: No

---

## [Author Response · Author response to Decision Letter 2]

22 Nov 2023

Reviewer #7: In the submitted manuscript “Prevalence of hepatitis B virus infection and its associated factors among students in N’Djamena, Chad”, PONE-D-22-22361R2, this study addressed the high prevalence of HBV infection among university students in Chad. HBV prevention, especially HBV immunization, should be considered in low-income countries.

However, there are some points that need to be clarified: 

1. the authors should mention inclusion criteria in the method part (e.g., sex, age, health status, etc

Thank you for this observation. We have revised the text accordingly, stating that students who accepted to participate in the study were enrolled consecutively irrespective of age, sex, other sociodemographic characteristics or risk factors. Line 105–108.

2. The data was collected using a questionnaire, was it a self-administered questionnaire or face-to-face interview? Please specify in data collection method. 

Thank you for your pertinent observation. The interviews were done during face-to-face meetings. Line 119–121.

3. The prevalence of HBV infection was 14.87%, what’s about 95%CI? Please specify.

14.87 is the prevalence of HBsAg in the population studied (95% CI = 13.9–21.7%). This confidence interval was added to the text. Line 191.

4. I would suggest using median age instead of mean age.

Thank you for this suggestion. Median age was 24 years. Line 182.

5. Considering p<0.30 in bivariate analysis, why did the authors select only five variables for multivariable logistic regression? (What about age, marital status, knowledge of HBV, history of drug injection, density of residence? there were also p<0.30.) 

Age was included in the analysis (age ≥26 was a protective factor). A previous reviewer and the editor suggested, with reason, to remove knowledge of HBV, which of course is a confounding factor. We had forgotten to remove it from the manuscript in the data analysis paragraph. Marital status, history of drug injection and density of residence were also included in the analysis, but were not significantly associated with HBs Ag positivity (see Table 3).

6. The authors did not ask about the history of HB immunization in the study participants. Although most likely did not benefit from vaccination against HBV in childhood as part of EPI, some may receive the vaccine in adult. We therefore can distinguish between who has protective or susceptible for HBV infection.

Thank you for your comment. Unfortunately, students were not asked about their vaccination status. We’ve touched upon this point in the discussion. Line 274–275

---

## [Editor Report · Decision Letter 3]

29 Nov 2023

PONE-D-22-22361R3Prevalence of hepatitis B virus infection and its associated factors among students in N'Djamena, Chad.PLOS ONE

Dear Dr. DEBSIKREO,

Thank you for submitting your manuscript to PLOS ONE. After careful consideration, we feel that it has merit but does not fully meet PLOS ONE’s publication criteria as it currently stands. Therefore, we invite you to submit a revised version of the manuscript that addresses the points raised during the review process.

**I commend the authors did a great job by incorporating reviewers' comments which substantively improved the paper. However, the paper doesn't until now rigorously align with Plos One's authors guidelines. Please note the following: **

**1. References: Plos One uses brackets; **

**2. In all Tables, add 1 decimal for percentages **

**3. When citing R, the date is unnecessary**

**Additionally, reviewer #2 has raised some comments to be integrated. I suggest to put these questions in the discussion section or alternatively, add a study limitations section.  **

We look forward to receiving your revised manuscript.

Kind regards,

Zacharie Tsala Dimbuene, Ph.D.

Academic Editor

PLOS ONE
---

## [Author Response · Author response to Decision Letter 3]

13 Jan 2024

Comments to the Author 

IComments to the Author 

I commend the authors did a great job by incorporating reviewers' comments which substantively improved the paper. However, the paper doesn't until now rigorously align with Plos One's authors guidelines. Please note the following:

1. References: Plos One uses brackets; 

Thank you for this observation, we've changed hooks

2. In all Tables, add 1 decimal for percentages

 Thank you for your comment, we have added 1 decimal for percentages. Table 1: line 185-186, Table 2:199-200 Table 3:212-213

3. When citing R, the date is unnecessary

 Thank you, we have deleted. Line 168

Additionally, reviewer #2 has raised some comments to be integrated. I suggest to put these questions in the discussion section or alternatively, add a study limitations section.

Thank you for this suggestion, we have added a section on the Strengths and limitations

---

## [Editor Report · Decision Letter 4]

18 Jan 2024

PONE-D-22-22361R4Title: Prevalence of hepatitis B virus infection and its associated factors among students in N'Djamena, Chad.PLOS ONE

Dear Dr. DEBSIKREO,

Thank you for submitting your manuscript to PLOS ONE. After careful consideration, we feel that it has merit but does not fully meet PLOS ONE’s publication criteria as it currently stands. Therefore, we invite you to submit a revised version of the manuscript that addresses the points raised during the review process.

We look forward to receiving your revised manuscript.

Kind regards,

Zacharie Tsala Dimbuene, Ph.D.

Academic Editor

PLOS ONE

Journal Requirements:

Additional Editor Comments:

Dear authors,

One of the reviewers suggested that final version should incorporate the following changes based on the manuscript evaluation:

"In the submitted manuscript “Prevalence of hepatitis B virus infection and its associated factors among students in N’Djamena, Chad”, PONE-D-22-22361R2, this study addressed the high prevalence of HBV infection among university students in Chad. HBV prevention, especially HBV immunization, should be considered in low-income countries.

However, there are some points that need to be clarified:

1. The authors should mention inclusion criteria in the method part (e.g., sex, age, health status, etc.).

2. The data was collected using a questionnaire, was it a self-administered questionnaire or face-to-face interview? Please specify in data collection method.

3. The prevalence of HBV infection was 14.87%, what’s about 95%CI? please specify.

4. I would suggest using median age instead of mean age.

5. Considering p<0.30 in bivariate analysis, why did the authors select only five variables for multivariable logistic regression? (What about age, marital status, knowledge of HBV, history of drug injection, density of residence? there were also p<0.30.)

6. The authors did not ask about the history of HB immunization in the study participants. Although most likely did not benefit from vaccination against HBV in childhood as part of EPI, some may receive the vaccine in adult. We therefore can distinguish between who has protective or susceptible for HBV infection."

Thank you

---

## [Author Response · Author response to Decision Letter 4]

1 Feb 2024

Response to reviewers

One of the reviewers suggested that final version should incorporate the following changes based on the manuscript evaluation:

"In the submitted manuscript “Prevalence of hepatitis B virus infection and its associated factors among students in N’Djamena, Chad”, PONE-D-22-22361R2, this study addressed the high prevalence of HBV infection among university students in Chad. HBV prevention, especially HBV immunization, should be considered in low-income countries.

However, there are some points that need to be clarified:

1. The authors should mention inclusion criteria in the method part (e.g., sex, age, health status, etc.).

Thank you for this observation. Students who accepted to participate in the study were enrolled consecutively irrespective of age, sex and other sociodemographic characteristics or risk factors line 105-109

2. The data was collected using a questionnaire, was it a self-administered questionnaire or face-to-face interview? Please specify in data collection method.

 Thank you for your pertinent observation. The data was it questionnaire face-to-face line 122-123

3. The prevalence of HBV infection was 14.87%, what’s about 95%CI? Please specify.

14.87 is the prevalence of HBsAg in the population studied (95% CI = 13.9-21.7%). Line 193

4. I would suggest using median age instead of mean age.

Thank you for this suggestion. Median age was 24 years. Line 185.

5. Considering p<0.30 in bivariate analysis, why did the authors select only five variables for multivariable logistic regression? (What about age, marital status, knowledge of HBV, history of drug injection, density of residence? there were also p<0.30.)

Age was included in the analysis (age>26 was a protective factor). A previous reviewer and the editor suggested, with reason, to remove knowledge of HBV, which of course is a confounding factor. We have forgotten to remove it from the manuscript in the data analysis paragraph. Marital status, history of drug injection and density of residence were also included in the analysis, but were not significantly associated with HBs Ag positivity (see table 3). Line 215

6. The authors did not ask about the history of HB immunization in the study participants. Although most likely did not benefit from vaccination against HBV in childhood as part of EPI, some may receive the vaccine in adult. We therefore can distinguish between who has protective or susceptible for HBV infection."

Thank you for your comment. Unfortunately, students were not asked about their vaccination status. Please see discussion — line 275—277

I commend the authors did a great job by incorporating reviewers' comments which substantively improved the paper. However, the paper doesn't until now rigorously align with Plos One's authors guidelines. Please note the following:

1. References: Plos One uses brackets; 

Thank you for this observation, we've changed hooks

2. In all Tables, add 1 decimal for percentages

 Thank you for your comment, we have added 1 decimal for percentages. Table 1: line 188-189, Table 2:203-204 Table 3:214-215

 3. When citing R, the date is unnecessary

 Thank you, we have deleted. Line 171

Additionally, reviewer #2 has raised some comments to be integrated. I suggest to put these questions in the discussion section or alternatively, add a study limitations section.

Thank you for this suggestion, we have added a section on the Strengths and limitations

I commend the authors did a great job by incorporating reviewers' comments which substantively improved the paper. However, the paper doesn't until now rigorously align with Plos One's authors guidelines. Please note the following:

1. References: Plos One uses brackets; 

Thank you for this observation, we've changed hooks

2. In all Tables, add 1 decimal for percentages

 Thank you for your comment, we have added 1 decimal for percentages. Table 1: line 188-189, Table 2:203-204 Table 3:214-215

3. When citing R, the date is unnecessary

 Thank you, we have deleted. Line 171

Additionally, reviewer #2 has raised some comments to be integrated. I suggest to put these questions in the discussion section or alternatively, add a study limitations section.

Thank you for this suggestion, we have added a section on the Strengths and limitations

However, there are some points that need to be clarified: 

1. the authors should mention inclusion criteria in the method part (e.g., sex, age, health status, etc

Thank you for this observation. We have revised the text accordingly, stating that students who accepted to participate in the study were enrolled consecutively irrespective of age, sex, other sociodemographic characteristics or risk factors. Line 105–109.

2. The data was collected using a questionnaire, was it a self-administered questionnaire or face-to-face interview? Please specify in data collection method. 

Thank you for your pertinent observation. The interviews were done during face-to-face meetings. Line 122–123.

3. The prevalence of HBV infection was 14.87%, what’s about 95%CI? Please specify.

14.87 is the prevalence of HBsAg in the population studied (95% CI = 13.9–21.7%). This confidence interval was added to the text. Line 195.

4. I would suggest using median age instead of mean age.

Thank you for this suggestion. Median age was 24 years. Line 185.

5. Considering p<0.30 in bivariate analysis, why did the authors select only five variables for multivariable logistic regression? (What about age, marital status, knowledge of HBV, history of drug injection, density of residence? there were also p<0.30.) 

Age was included in the analysis (age ≥26 was a protective factor). A previous reviewer and the editor suggested, with reason, to remove knowledge of HBV, which of course is a confounding factor. We had forgotten to remove it from the manuscript in the data analysis paragraph. Marital status, history of drug injection and density of residence were also included in the analysis, but were not significantly associated with HBs Ag positivity (see Table 3).

6. The authors did not ask about the history of HB immunization in the study participants. Although most likely did not benefit from vaccination against HBV in childhood as part of EPI, some may receive the vaccine in adult. We therefore can distinguish between who has protective or susceptible for HBV infection.

Thank you for your comment. Unfortunately, students were not asked about their vaccination status. We’ve touched upon this point in the discussion. Line 274–275

---

## [Editor Report · Decision Letter 5]

5 Feb 2024

PONE-D-22-22361R5Titre : Prévalence de l'infection par le virus de l'hépatite B et ses facteurs associés chez les étudiants de N'Djamena, Tchad.PLOS ONE

Dear Dr. *NALDA DEBSIKREO*,

Thank you for submitting your manuscript to PLOS ONE. After careful consideration, we feel that it has merit but does not fully meet PLOS ONE’s publication criteria as it currently stands. Therefore, we invite you to submit a revised version of the manuscript that addresses the points raised during the review process.

**Dear authors, please note the following: **1. I have extensively edited the manuscripts. Kindly incorporate these changes 2. I am a bit concerned with Table 3. In this Table, some variables do not have confidence intervals in "multivariate analyses". I can understand that some variables were not included in multivariate analyses. However, doing this makes the Table a bit cumbersome to understand. I suggest that you separate bivariate Table and multivariate Table (in which ONLY variables retained in multivariate analyses are included. That way, each variable (or categories of variables) will have a confidence interval. 

We look forward to receiving your revised manuscript.

Kind regards,

Zacharie Tsala Dimbuene, Ph.D.

Academic Editor

PLOS ONE
---

## [Author Response · Author response to Decision Letter 5]

28 Feb 2024

Dear authors, please note the following: 

1. I have extensively edited the manuscripts. Kindly incorporate these changes 

We have been able to incorporate changes as requested by the Editor. Line: 18-20, 49, 55-57; 62; 64; 72; 75;95; 97; 116; 134; 142; 199; 230-231; 235; 238; 

2. I am a bit concerned with Table 3. In this Table, some variables do not have confidence intervals in "multivariate analyses". I can understand that some variables were not included in multivariate analyses. However, doing this makes the Table a bit cumbersome to understand. I suggest that you separate bivariate Table and multivariate Table (in which ONLY variables retained in multivariate analyses are included. That way, each variable (or categories of variables) will have a confidence interval. 

Thank you for this suggestion, we have table 3 in 2 tables. Table 3: line 216 and table 4: 228

---

## [Editor Report · Decision Letter 6]

6 Mar 2024

Title: Prevalence of hepatitis B virus infection and its associated factors among students in N'Djamena, Chad

PONE-D-22-22361R6

Dear NALDA DEBSIKREO,

We’re pleased to inform you that your manuscript has been judged scientifically suitable for publication and will be formally accepted for publication once it meets all outstanding technical requirements.

Kind regards,

Zacharie Tsala Dimbuene, Ph.D.

Academic Editor

PLOS ONE
---

## [Editor Report · Acceptance letter]

11 Mar 2024

PONE-D-22-22361R6 

PLOS ONE

Dear Dr. Debsikréo, 

I'm pleased to inform you that your manuscript has been deemed suitable for publication in PLOS ONE. Congratulations! Your manuscript is now being handed over to our production team.

Kind regards, 

on behalf of

Prof. Zacharie Tsala Dimbuene 

Academic Editor

PLOS ONE